# Optimization of Surface Preparation and Painting Processes for Railway and Automotive Steel Sheets

**Szabolcs Szalai** [1], **Brigitta Fruzsina Szívós** [1], **Dmytro Kurhan** [2], **Attila Németh** [1,*], **Mykola Sysyn** [3] and **Szabolcs Fischer** [1,*]

1 Central Campus Győr, Széchenyi István University, H-9026 Győr, Hungary
2 Department of Transport Infrastructure, Ukrainian State University of Science and Technologies, UA-49005 Dnipro, Ukraine
3 Department of Planning and Design of Railway Infrastructure, Technical University Dresden, D-01069 Dresden, Germany
* Correspondence: nemeth.attila@sze.hu (A.N.); fischersz@sze.hu (S.F.); Tel.: +36-(96)-613-544 (S.F.)

**Abstract:** The article deals with DIC (Digital Image Correlation) tests on steel plates used in the automotive and railway industries, as well as in the construction industry. The most critical part of DIC tests is the quality of proper surface preparation, painting, and random patterns. The paint mediates the deformation of the optical systems, and its quality is paramount. The authors' goal in this research is to determine the optimal dye–cleaning–drying time parameters for DIC studies. Commercially available surface preparation and cleaning agents were tested alongside commercially available spray paints. Standard and specific qualification procedures were applied for the measurements. Once the appropriate parameters were determined, the results were validated and qualified by GOM ARAMIS tests. Based on the results, DIC measurements can be performed with higher accuracy and safety in laboratorial and industrial conditions, compared to the traditional deformation measurements executed by dial gauges or linear variable differential transformers.

**Keywords:** speckle pattern; DIC; painting; surface cleaning; GOM ARAMIS; automotive steel; specimen preparation; Erichsen cupping test





## 1. Introduction

Several methods for measuring deformations and displacements are known from the literature and metrological practice. Traditional methods of measurement include, among others, strain gauges, stamps, and inductive transducers. Unfortunately, these measurement methods require much preparation and expensive tools and instruments.

In this research, the Digital Image Correlation (DIC) techniques presented are non-contact techniques that allow the investigation of the total displacement and strain field of the parts of the specimen visible to the camera using common standard tools, with post-processing of the results [1]. For these measurements, it is not necessary to know the location of the failure in advance since the measurement results are obtained for the whole specimen and are processed afterward [1].

The DIC method can also be used in mechanical, civil, electric, and other engineering research, projects, and studies. They can be related to railways [2–6], highways [7–10], air transport [11,12], shipping [13], astronautics [14,15], and so on. In this research, the authors applied an approach regarding railway and automotive vehicles and their car body sheets.

The DIC as a testing technique was first developed in the 1980s by a group of researchers at the University of South Carolina, and over the next three decades, it has evolved and become widely used. Thanks to its many advantages, such as its easy experimental set-up, variability, and widely adjustable time and spatial resolution, the DIC technique has become widely accepted as a helpful tool for measuring deformation [16].

Nowadays, researchers also prefer this technology, as illustrated in the diagram in Figure 1. The graph shows the number of publications on Google Scholar between 1990 and 2022, where the dynamic growth of DIC technology is visible. The ESPI (Electronic Speckle Pattern Interferometry) is the so-called traditional technology in this field.

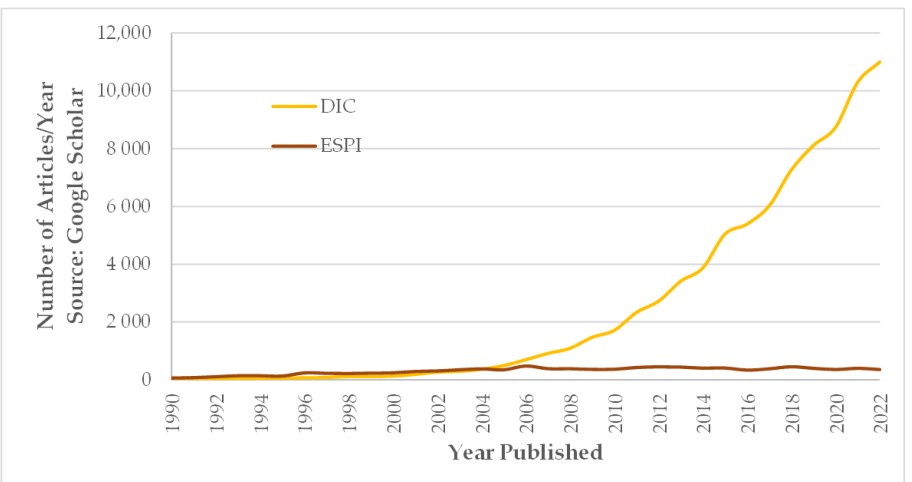

**Figure 1.** Increasing use of the DIC technique in scientific publications (on the basis of Google Scholar).

Based on these results, the DIC method is one of the most critical measurement equipment in the experimental engineering and research community, and their applicability will continue to grow [17,18]. The main strength of DIC is that its principle is valid on microscopic (millimeter scale test specimens) and macroscopic (meter scale test) scales [17,18].

In addition to the range of sizes that can be used, it is interesting to note that the technology has not only been applied to testing metallic materials but is nowadays also applied for testing polymers, composites, and even biological materials [18]. In addition to conventional test specimens, it is also used in complex components, machinery, building components, biomechanics, and the medical sciences [18].

Referring to Figure 1, 164 and 20 papers have been published—in 2022 and 2023, respectively—indexed in the Scopus database, with the titles containing DIC. According to these searching criteria, there are 1054 documents between 2001 and 2023. The top-rated areas are engineering, material sciences, physics, and astronomy. If one would like to overview the citations in Scopus, the most cited papers are the following [19–23]. From the newest literature, the following papers are worth mentioning [24–28]. Therefore, at this stage, it can be concluded that the DIC techniques are and will be significant parts of research and development in the 21st century.

*1.1. Imaging in DIC Processing*

As for the recording process, all images were taken with analog cameras in the early years, which required digitizing hardware. The early digital cameras were undoubtedly an improvement, although the digital data later had to be transferred to a mainframe computer for processing to analyze the images. As processing speeds increased and storage costs decreased, mainframes were replaced by PCs (i.e., personal computers), which gave greater flexibility in using DIC methods.

Today, an experiment typically produces a few thousand images per camera. The increase in helpful information has improved our understanding of physical phenomena. However, such a large amount of information requires algorithms that can efficiently and accurately analyze the images to improve throughput and thus requires continuous improvement [1,18].

In order to determine the surface deformation, surface images of the test specimen in different configurations are captured and stored on a computer for post-processing. The two-dimensional digital image correlation (2D-DIC) technique can be used to determine the

main deformations. However, since 2D-DIC uses a single camera for image acquisition, the method is limited to in-plane case studies and planar motion. Out-of-plane displacements introduce errors that affect the accuracy of the results.

Since we cannot determine the size of objects using this method, we must assume that the object is flat, parallel, and at a constant distance from the visual sensor. If the object's surface is not planar or a 3D deformation occurs after loading, the 2D-DIC method is no longer applicable. To measure displacements and deformations in three-dimensional space, the 3D-DIC technique must be applied.

The 3D-DIC technique uses two or more cameras to simultaneously capture an image of the examined object from different angles [17,18,29–31]. Two imaging sensors already provide enough information to perceive the environment in three dimensions. Not only 3D objects but also moving objects and objects subject to large deformations can be introduced using the 3D-DIC technique [17,18,29–31].

After capturing the images with one or more digital cameras, the next stage of the test is image analysis (evaluation) in a special computer program. The DIC system is based on the comparison of digital images. Mathematical correlation analysis is used to analyze digital images captured before deformation ('reference image') and during the deformation process ('deformed image').

The program can display the bulk deformation by comparing digital images, determining the displacement field, and mapping its distribution. The surface of the test piece, as the carrier of the deformation information, should have a random intensity spot pattern. By processing the digital images, the software recognizes the structure of the stochastic pattern and assigns coordinates to the pixels of the image [17,29,31,32]. The object's deformation is measured over overlapping image patches of a specific size. Smaller image segments (rather than individual pixels) are used because the part with the larger grayscale difference can be identified from the rest of the distorted image [17,29,31,32].

### 1.2. The Importance of Patterns and Dyes in DIC Studies

DIC measurements cannot be performed on specimens without samples, as the painting pattern has an important impact on the accuracy and precision of the displacements detected by DIC. Presumably, image-based DIC techniques can be extended to any test sample, provided that the captured digital images have adequate and stable intensity variations and show unique matches with the physical points on the test sample surface [1,18,31,33,34].

In general, a good paint sample should meet the following quality requirements. First and foremost, a high contrast is needed, i.e., a pattern with a varied intensity of greyscale. An important aspect is randomness, i.e., a non-periodic and non-repetitive pattern, to facilitate the mapping of the overall spatial displacement [18,34]. Isotropy should also be mentioned, as there should be no obvious directionality in the pattern. However, one of the most critical aspects is stability, as the spot pattern must adhere well to the surface of the specimen to ensure that the deformation is adequate [18,34].

According to Phillip Reu, four essential characteristics of a spot pattern need to be addressed in more detail: spot size, contrast, the overlap between spots, and spot density. The combination of these four characteristics helps to determine the correct spot pattern. The size of the speckles needs to be determined from the relationship between the resolution of the camera used for DIC and the area of the specimen under testing. The size of the pixels in millimeters can be calculated from these two data. The optimum spot size is usually in the range of 3–5 pixels, as smaller pixels are more difficult to detect during digitization.

However, it is not only the size of the spots that is important, as the minimum size of three pixels is also a significant factor. If the overlap of the speckle pattern is inadequate, a large amount of unnecessary noise can be generated during digitization when taking images. For this reason, avoiding overlapping when applying the spot pattern is very important, as it can negatively affect the measurement results.

Overlapping also raises the issue of patch density, as a pattern applied too often can lead to measurement errors. A printed pattern can be used to avoid it, but this option is

not always available. Consequently, when applying the pattern by spray painting, care must be taken to avoid 'overspray' of paint onto the specimen surface. When creating a painting pattern, concentrating on the size and density of the paints can lead to the mistake of having a contrast of dots that is not of sufficient quality to make measurements. The contrast of a black spot pattern applied to a typically white ground surface helps to identify specific areas on the specimens [34].

With the right paint and lighting, the contrast can be maximized and simultaneously reduce noise levels, an inevitable aspect [35–38].

During the measurement, the task is to compare a series of images of a specimen coated with a unique pattern and to search for and trace identical points throughout the images. However, searching for a single pixel between two images may not yield results, as the color and/or intensity value of a pixel may be the same as that of several pixels in the reference image. A solution is to identify a small area (evaluation window) in the vicinity of a pixel between images.

The seek evaluation window is usually square, identifiable through the entire sequence of images due to the unique pattern captured during the test specimen preparation. This deformation process where the detection window can follow the pattern is shown in Figure 2 [39–41].

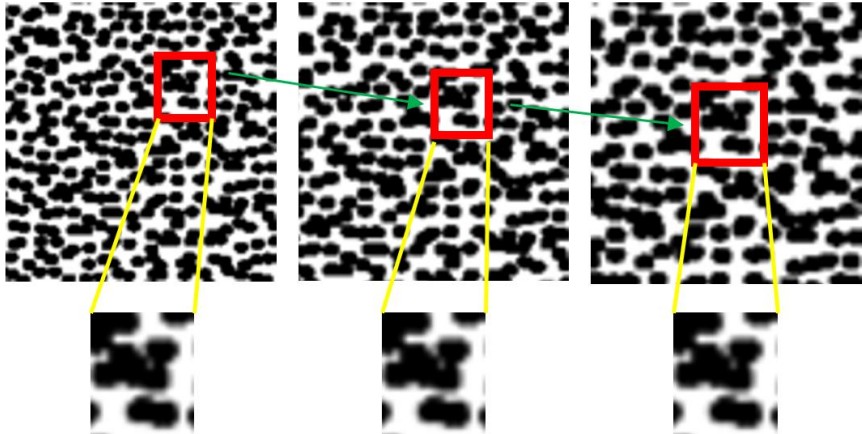

**Figure 2.** Deformation of the evaluation window during the measurement (on the basis of [30]).

There are several methodologies to find a region around a pixel in another image. The two most common are the normalized cross-correlation and normalized least squares method. In order to achieve sufficient accuracy, the deformation of the evaluation windows must be taken into account, for which some form of iteration is usually used [39,41], the steps of which are shown in Figure 3.

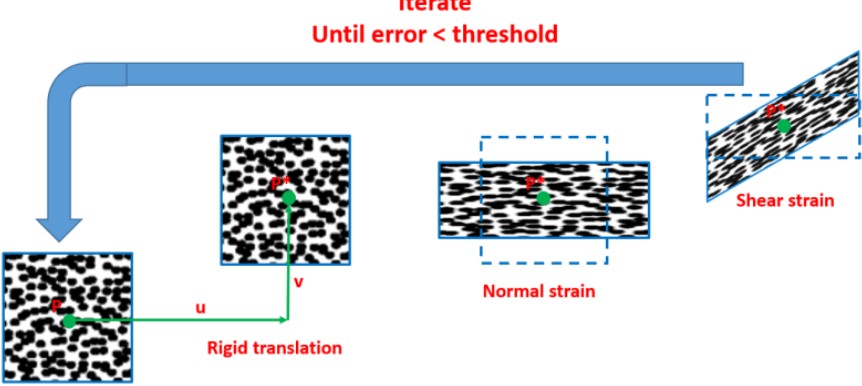

**Figure 3.** The steps of the iteration (on the basis of [41]).

### 1.3. DIC Solutions in Formability Studies

More recently, Murat Aydin et al. [42] used the DIC technique to measure the height of the Erichsen deep-drawn specimen in their study. They operated the DIC stereo camera and the Erichsen test set-up together to determine the height and thickness distribution of the recessed specimen. However, the DIC requires a high-resolution camera because the DIC has to detect small points on the surface of the specimen. For the same reason, as a stereo camera, more than one camera must be used to measure the height of the cup [17].

In the forming of the experiments, a cupping test known as the Erichsen test was used to stretch flat specimens to fracture to determine the ductile behavior of sheet metal materials. The deformation test consisted of biaxially stretching a square flat specimen clamped between punches. For the Erichsen test, specimens of a specific size were prepared and simply placed between the dies for clamping. Each specimen had to be prepared for the test [42].

At first, the dirt was removed from the surface of the pieces and then cleaned with acetone. Preparation continued with the painting phase, where a base coat of white paint was applied to the surface of the pieces, followed by a random paint pattern using black paint. All experiments were conducted at room temperature and in a laboratory environment [42]. Specimens were illuminated with white light and layered between dies to avoid shadow and focus errors. Before starting the test, it was necessary to carry out the DIC calibration process to obtain accurate and precise deformation results [42].

Murat Aydin et al. [42] chose Vic3D software for image processing and deformation field calculation among the many DIC software available on the market. DIC measurements gave acceptable and adequate results compared to height measurements. The physical instrumental measurements were slightly above the DIC measurements, as during the forming experiments, the crosshead of the manually operated hydraulic press moved slightly further after stopping [17,42].

Thanapat Sangkharat et al. [17] focus on determining the anisotropic property using a biaxial force test. The Erichsen test is a biaxial, i.e., two-axial force test method, and thus, this study chose this test method to shape the specimens. The anisotropic property was calculated using the digital image processing method.

In addition, this study [17] uses only one camera to determine the anisotropic properties of the test specimen, in which the camera was mounted on top of the test machine, and an LED circular light was used to help provide adequate illumination to the camera. Illumination from the LED circular light was direct, thus the light beams were directly incident on the object, and then the beams were reflected to the camera.

This study [17] uses Open Computer Vision (OpenCV), a powerful open-source computer program. For the study, rolled tin plates were used as specimens and placed at different angles depending on the rolling direction for the Erichsen test. From the image recordings, it can be concluded that the angular direction of the specimen affects the direction of cracking in the Erichsen test. This means that the Erichsen test can determine the rolling direction of the plates since cracks are formed in the direction perpendicular to the rolling direction, and therefore, the test can determine the anisotropy of the metal plate.

In an experiment by Fabian S. Sorce et al. [43], the Erichsen test was performed to investigate the resistance of coatings to failure. In this test, a hemispherical cupping machine is used to observe the failure of the coating during the forming process.

In industry, tests like this are typically evaluated from a quality control perspective with a yes/no rating depending on whether the coating meets the deformation requirements. Coating formability is essential because of the manufacturing process of coil-coated metal products. Coatings are used primarily for protection, good appearance, and high quality. Resistance to cracking is, therefore, an essential requirement for coatings, together with good adhesion to the primer and substrate.

In this experiment, the surface deformations in the Erichsen test are demonstrated using a finite element model. The predicted deformations are compared with those determined experimentally using DIC. For each result, deformation characteristics were

evaluated, allowing the prediction of the expected location of coating failure. The experiment was performed using GOM's ARAMIS system, and the results were analyzed using GOM Correlate Professional software. In the experiment by Sorce et al. [43], in all of the Erichsen tests, the steel plate failed before coating, indicating that the ductility of the coating was better than that of the steel [43].

Considering the results of the studies mentioned above, it can be stated that significant results can be achieved by combining the Erichsen test with digital image processing and image correlation.

### 1.4. The Novelty of the Current Study and the Structure of the Paper

Based on the literature review shown in Sections 1.1–1.3, the authors have decided to investigate the suitability and applicability of a combination of commercially available cleaning agents and paint sprays on five different types of steel plates for DIC-based optical deformation measurements combined with Erichsen cupping tests. For the DIC measurements, a GOM ARAMIS 5M system was used. The qualification of the paintings started with a series of measurements according to international standards. These were the so-called pre-filters for the Erichsen cupping test. Until now, this detailed test series has not yet been executed related to the surface preparation and painting processes for railway and automotive steel sheets considering DIC measurements. This paper aimed to fill this research gap partially.

The structure of the paper is the following: in Section 2, the applied materials and methods are detailed. Section 3 deals with the results, point-by-point, in accordance with Section 2.2. Finally, in the end, Section 4 contains the derived conclusions.

The laboratory experiments were performed by wet paint film thickness tests (see Sections 2.2.1 and 3.1), touch tests (Sections 2.2.2 and 3.2), print-free tests (Sections 2.2.3 and 3.3), cross-cut tests (Sections 2.2.4 and 3.4), bending tests (Sections 2.2.5 and 3.5), Erichsen tests with DIC evaluation and traditional dial gauge measurement (Sections 2.2.6 and 2.2.7 as well as Sections 3.6 and 3.7), as well as durability tests (Section 3.8).

## 2. Materials and Methods

The main focus of the research was to investigate commercially available surface preparation and cleaning materials and spray paints used in industrial and laboratorial research processes. The choice of surface preparation materials was also based on the manufacturer's recommendations.

In addition to easy accessibility, the selection of paints was based on the criteria of easy spray application and ease of use, preferably for indoor use. Twelve surface cleaners, eleven paints, and five steel sheet materials were tested during the tests.

Because of the fact that the authors have more than 10 years of experience in DIC measurements, the preparation of adequate, appropriate, and the same painting technology, as well as the same speckle pattern, for all the specimens was guaranteed.

### 2.1. Materials Applied in the Study

The following surface preparation, surface cleaning agents, and liquids were used for the tests (Table 1 and Figure 4).

The criteria for the selection of paints have been described earlier. The aim was to test white paints with good hiding power and those readily available commercially in spray form. The research is limited to white paints because the DIC system tests for accidental paint patterns are most often made with black paints. In addition, white provides the best contrast to black paint, so the primer paint should be of this color.

The reason for the spray formulation was that measurements often have to be carried out under non-laboratory conditions or in inadequately equipped laboratories where airbrush spray painting procedures cannot be used. Therefore, another criterion for selection was that the paint should preferably be suitable for indoor use and should be able to be

applied with the use of personal respiratory protection equipment. The following table summarizes the selected paints (Table 2 and Figure 5).

**Table 1.** Surface preparation materials and processes.

| ID. | Name | Brand |
|---|---|---|
| SC0 | wipe dry cleaning | Tork 652100 |
| SC1 | brake cleaner | Engelnert Strauss |
| SC2 | acetone UN1090 | Styro-Flow |
| SC3 | Brigéciol D-3 | Kemobil |
| SC4 | dilutive 513 | Egrokorr Izofix |
| SC5 | contact cleaner | Chip Medikémia |
| SC6 | 9973 engine washer | Mannol |
| SC7 | H-100 synthetic diluent | Supralux |
| SC8 | nitro-dilution UN1263 | Styro-Flow |
| SC9 | isopropyl alcohol | Győrlakk |
| SC10 | cold degreaser | Welldone |
| SC11 | detergent water | Ludwik |

**Table 2.** List of paints used in the research.

| ID. | Name | Manufacturer | Brightness | Color | Dispensing [mL] | Base (Min 40%) |
|---|---|---|---|---|---|---|
| P1 | Acrylic Primer | Maestro | mat | RAL9003 | 400 | Acetone + Xilol |
| P2 | Acrylic Mat Primer | Maestro | mat | RAL9003 | 400 | Acetone |
| P3 | High Temp. Paint | Maestro | silk | RAL9003 | 400 | Acetone + Xilol |
| P4 | Heat Resistant | Motip | mat | 04036 (white) | 400 | Silicone Resin |
| P5 | Spray Putty | Motip | mat | 04062 (beige) | 400 | Acrylic resin |
| P6 | Acryl | Prisma Color | silk | RAL9010 | 400 | Acetone + Ethyl acetate |
| P7 | Acryl | Prisma Color | silk | RAL9016 | 400 | Acetone + Ethyl acetate |
| P8 | Radiator | Prisma Color | silk | 91152 (white) | 400 | Acetone + Ethyl acetate |
| P9 | Aqua Eco+ | Dupli-Color | mat | RAL9010 | 350 | Water + Acetone |
| P10 | Chalk Finish Broken white | Pinty Plus | mat | CK788 | 400 | Water + Acetone |
| P11 | HB Body 950 | HB Body | mat | White | 400 | Caucuses + Syntetic Resin |

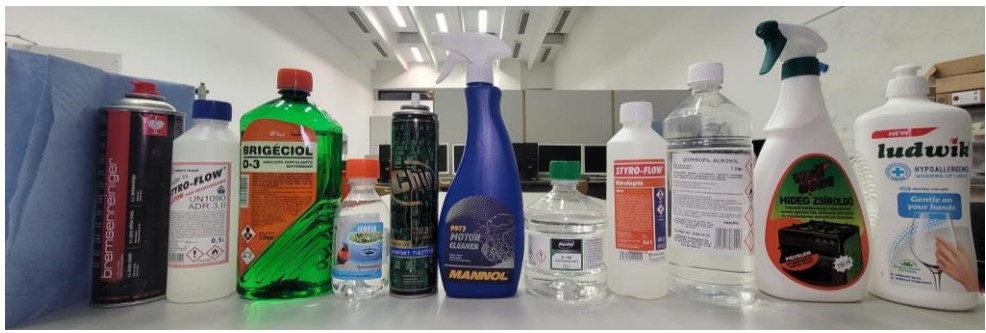

**Figure 4.** The applied cleaning products.

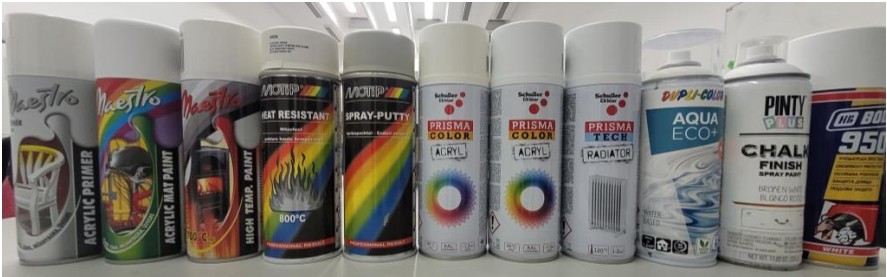

**Figure 5.** The applied paints.

The steel plates selected for the research were those used in the automotive industry and the manufacturing of railway vehicle cabinets. Therefore, it was necessary to use body materials with the composition and surface coating used in industrial practice. Regarding plate thicknesses, commonly occurring sizes were also selected in line with material stock availability and stockholding. The following table shows the five materials selected according to these criteria (Table 3).

**Table 3.** Steel sheet materials used for painting tests.

| ID. | Material Quality | Plate Thickness [mm] EN10131 |
|---|---|---|
| S1 | X5CrNi18-10 | 2.00 |
| S2 | DP600 | 1.00 |
| S3 | TRIP780 | 0.75 |
| S4 | DX54 | 1.00 |
| S5 | DX51D | 1.50 |

*2.2. Measurement Methods and Instruments Used in the Research*

This chapter describes all the test methods and tools used in the research and the standards used or related to them. There may be deviations from the relevant standard to suit the purpose of the research, but these will be justified in the relevant chapter.

The test pieces were cut to a uniform size of 90 × 90 mm using a pair of board shears. This size was used for all the measurement procedures in the research, so it was not necessary to use different specimen geometries. Before starting the tests, the specimens were cleaned with the cleaning agents listed in Table 1. All material types were cleaned with all cleaning agents.

Using rubber gloves was mandatory during the cleaning process to avoid contamination of the test piece after washing. Cleaning agents were applied uniformly to the surfaces using Tork 652,100 wipes. The advantage of the chosen wipe is that its material does not leave particles on the cleaned surface, unlike other paper wipes.

After cleaning, the test specimens were prepared with all eleven different paints according to the test procedure. In the present study, following the standards, the tests were carried out at a temperature of 22 °C +/−1 °C and a relative humidity of 50 RH% +/−5 RH%. In the following sub-sections, these procedures are described.

2.2.1. Wet Paint Film Thickness Measurement

The thickness of the paints measured in the wet state was determined during the test. Therefore, when applying the paint, care must be taken to ensure that it is applied in an even layer with uniform coverage. The test was carried out using an Elcometer 112Al wet film comb (Figure 6) and is in accordance with ISO 2808:2020—Paints and varnishes; Determination of film thickness standard test [44].

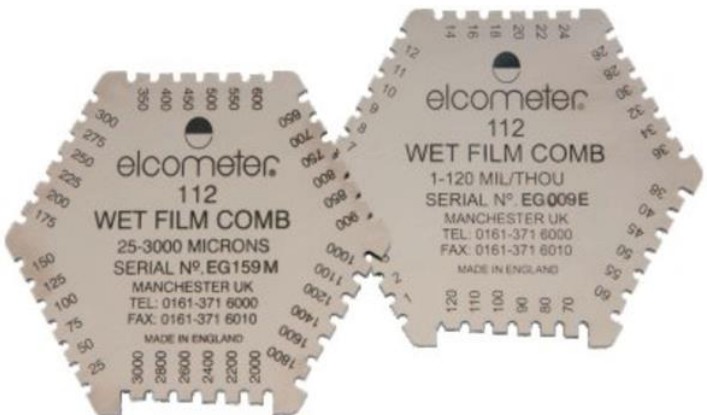

**Figure 6.** Elcometer 112Al.

Immediately after painting, the combs of the testing tool are pressed into the wet paint, and the thickness value in microns (i.e., $10^{-6}$ m) is read off of the side of the tool by highlighting it. The measurement should be taken three times per test specimen, and the results averaged. The measuring range of the instrument is 25–3000 μm.

### 2.2.2. Touch Test

The following test procedure is a subjective measurement method. Its purpose is to determine the time after the test specimens are mobile, i.e., they leave no tactile trace. It is necessary because, during the preparation of the specimens for a DIC test, the specimens must be dried in a different place where the paint is applied. Therefore, this test can determine the shortest time the paint has dried to the point where it does not leave a stain due to a small force. Other tests cannot be started within this time.

For the test, the test specimens shall be inspected every five minutes after painting, where the painted surface shall be touched by hand and checked for traces of paint. The use of rubber gloves is mandatory for the test. If the paint leaves a mark, the test shall be repeated after 5 min. If there is no stain, the test sample has passed the test.

### 2.2.3. Print-Free Test

The test determines the catch-proof drying of the paint. Compared to the previous test, this test shows how long the paint can be loaded, i.e., the paint is not yet solid but does not stain or come off the surface when weight is applied. The test is based on ISO 9117-6:2013—Paints and varnishes, Drying tests—Part 6: Print-free test; and ISO 9117-5:2013—Paints and varnishes, Drying tests—Part 5: Modified Bandow; Aligned to Wolff test standards [45,46].

The two standards are used because DIC tests should not be performed with paint that has dried in the wet state, based on literature data and practical experience. After all, if the paint is no longer elastic, it cannot transmit the deformation. Some deviation from the two relevant standards is therefore necessary. The modification only affects the weights; the maximum load is 1000 g. At this weight, the paint is sufficiently solid and stable; it does not separate when the pieces are moved or gripped, but it has the expected elasticity, i.e., it has not yet hardened. The weights used for the print-free test are shown in Figure 7.

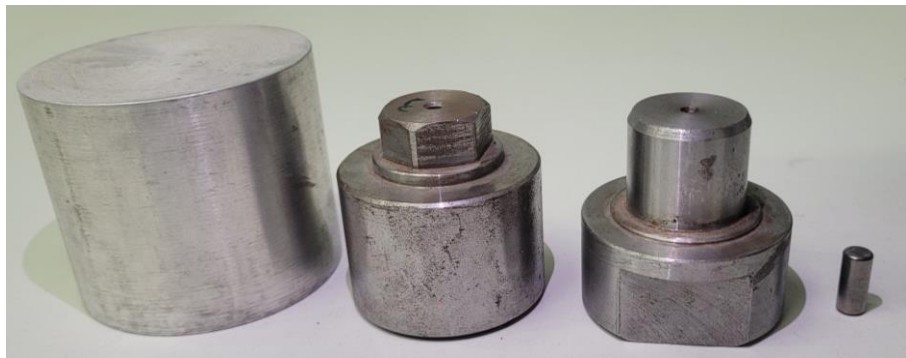

**Figure 7.** Weights applied for the print-free test.

During the test, a 26 mm diameter circular paper (80 g/m$^2$) is placed on the painted surface, followed by a 22 mm diameter 5 mm thick rubber element (hardness 50 IRHD +/−3 IRHD, ISO7619 [47]). Next, weights are placed on this rubber element, with a load of 20 g, 200 g, 500 g, and 1000 g in weight units. Each of these weights shall be loaded for 60 s, and after this time, the test piece shall be tapped against a 30 mm thick wooden surface from a height of at least 20 mm.

If the paper falls off, the next weight follows. If it sticks to the ink, the dip must be repeated after 15 min. A given ink passes the test if it passes the 1000 g load. The test determines the time after which the test specimen can be loaded. The test shows the load

capacity but does not show whether the paint adheres well to the surface to be tested, so further testing is required.

### 2.2.4. Cross-Cut Test

At this stage, a standard grid-cutting test is performed according to ISO 2409:2013—Paints and varnishes [48]; Cross-cut test according to the Elcometer 107 Cross Hatch cutter with 6 × 2 mm and 6 × 1 mm test heads (Figure 8). The test determines whether the paint has sufficient adhesion to the surface after the catch drying, i.e., whether it has a mediating deformation capacity.

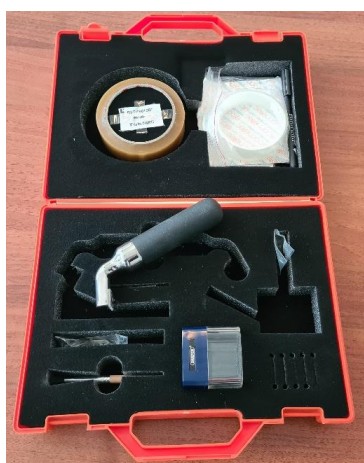

**Figure 8.** Elcometer 107 Cross-hatch cutter.

The tests were to be carried out 6, 12, and 24 h after the print-free drying times. A painting/cleaning method that shows poor results after 24 h is no longer applicable, as the manufacturer's recommendations indicate that the paint will cure and cease to perform its function after this time. Depending on the thickness results of the wet paint film, a 1 mm or 2 mm cutting head was used for the tests.

The cutting and measuring were done according to the standard, and the result was evaluated according to the standard. During the test, 6 to 6 parallel scratches were made perpendicular to each other with a cutting head of the thickness of the ply, and the layers loosened by cutting were removed with an adhesive strip of the adhesive strength specified in the standard. When evaluating the samples, it shall be observed whether there is any evidence of paint peeling along the grid cuts after the adhesive strip has been peeled off. As the goal is not to have fully cured paint, a rating of 2 is considered a good result, in addition to 0 and 1, which is slightly different from the standard (Table 4).

**Table 4.** Classification of the cross-cut test (on the basis of [48]).

| Classification of Adhesion Test Results | |
|---|---|
| **Classification** | **Percent area removed**<br>**It has to be determined by the surface of cross-cut area from which**<br>**flaking has occurred for six parallel cuts and adhesion range by percent** |
| 58 | 0%<br>None |
| 48 | Less than 5% |
| 38 | 5–15% |
| 28 | 15–35% |
| 18 | 35–65% |
| 08 | Greater than 65% |

### 2.2.5. Bending Test

In the bending test, the painted sheets are tested with a 20 mm diameter bending roller (Figure 9). The purpose of the measurement is to check the ductility and adhesion of the paint. The test is based on ISO 1519:2011—Paints and varnishes; Bend test (cylindrical mandrel) and ISO 1518:2000—Paints and varnishes Scratch test standard [49,50]. Preliminary experiments in this research have shown that there is a condition where the paint layer does not crack after bending, i.e., it is adequate, but it has separated from the surface of the sheet. Thus, by visual inspection, defective painting cannot be filtered out, which does not adequately convey the deformation parameters of the test specimen to be measured. The bend test combined with the scratch test helps to detect this phenomenon.

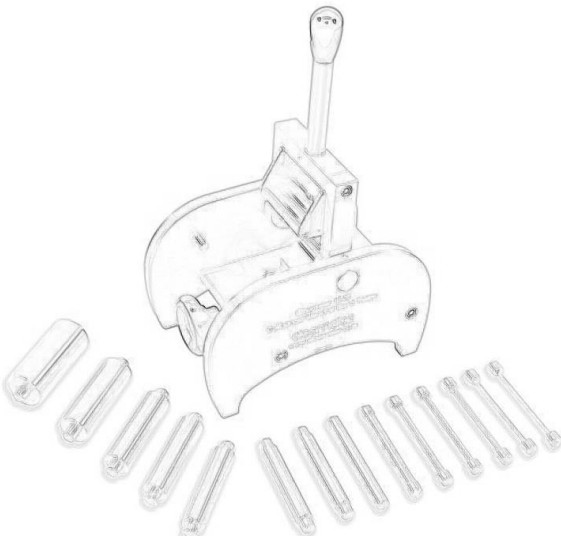

**Figure 9.** Bending device.

For the test, the painted 90 × 90 mm test plate should be scratched in the middle with a scratching needle so that the painting is cut through completely, if possible. The 20 mm bending roller shall then be used to bend the specimen 180°. When evaluating the results, the following classification criteria shall be used:

1. after bending, the paint does not separate from the plate along the cut;
2. after bending, flaking and slight cracking are observed along the cut;
3. after bending, there is no adhesion or contact along the cut, or anywhere the paint separates from the plate.

The modified bend test shall be carried out 6, 12, and 24 h after the paint has been applied and after the technological time for the catch drying, taking into account the results of the grid cut test.

### 2.2.6. Erichsen Test with DIC Evaluation

The Erichsen test plays a vital role in research. Painting/cleaning methods that score well in the pre-tests are checked and validated by this test. The test and the tools used comply with the ISO 20482:2013 Metallic materials—Sheet and strip—Erichsen cupping test standard [51]. The measurement was made on a hydraulic Erichsen tester with a 20 mm diameter test cylinder with a fixed binder.

For fixed binder force, the die and the binder must be pressed together so that the sample sheet cannot slip between them, and the material can only stretch. The test lasts until the first crack appears; the crack punch displacement is recorded at the end of the process. During the measurement, the GOM ARAMIS 5M DIC system, placed above the test area, records the deformation of the specimen (Figure 10). When the results are evaluated, the plate thinning on the cracked specimen is extracted and compared with the results

of the plate thickness measurement. In addition to these data, the punch displacement is extracted from the DIC measurement and compared with the data measured on the Erichsen machine.

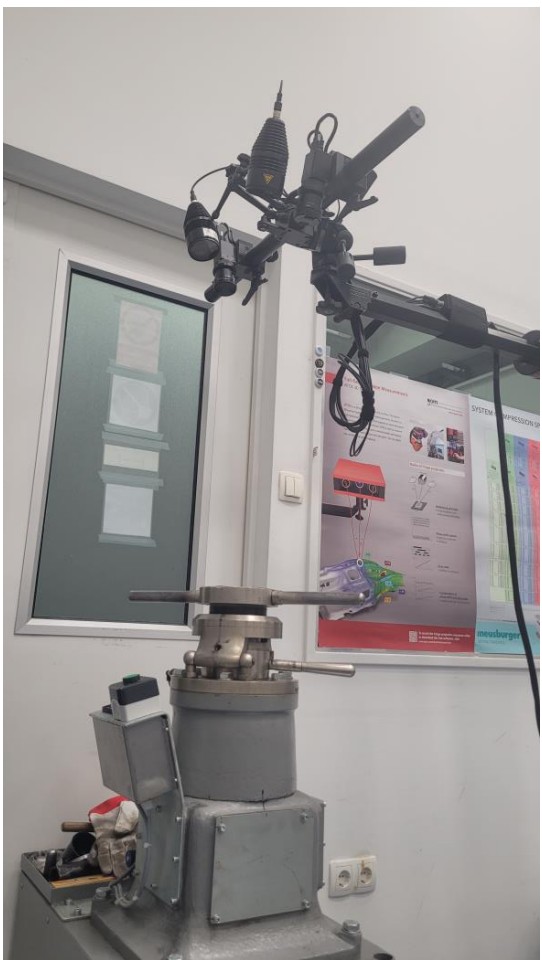

**Figure 10.** Erichsen testing equipment with GOM ARAMIS 5M DIC system (own photo).

If the results show a deviation of less than 10%, the painting/cleaning process is rated as satisfactory.

In addition to the measurement of deformation, the painted surface is visually inspected and classified into the following three categories:

1. the paint does not separate from the plate along the tear after pulling;
2. after pulling, flaking and slight cracking are observed along the tear;
3. after pulling, the paint separates from the disc along the tear or anywhere else, with no adhesion or contact.

### 2.2.7. Plate Thickness Test

During the plate thickness test, the thinning of the specimens is measured, and the results are compared with the results of the DIC test. Thus, the results are used to validate the DIC measurement as well as to qualify the different painting/cleaning procedures. For the test, the shaped specimens are measured at their apex. A dial gauge with an accuracy of 0.01 mm is used for the test. The measuring gauge type is Mitutoyo ID-S1012XB, 16037838, measuring range of 12.7–0.01 mm (Figure 11). The error of the gauge and the thickness of the paint layer must be taken into account when evaluating the results. The measuring error of the dial gauge bars is 0.02 mm, so this value should be considered when evaluating the measurements.

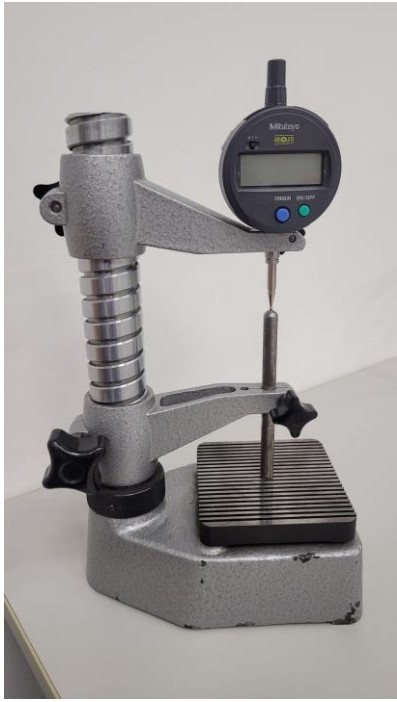

**Figure 11.** Dial gauge with a stand.

### 2.2.8. Durability Test

A longer-term usability test was conducted to complement the results of the DIC study. For the tests, the selected combination of steel sheet materials with cleaning agents and spray paints, which are the best based on previous measurements, were considered to examine further. The Erichsen tests were measured after a waiting period of 24 h for one week and again after two weeks. The study aimed to determine the more extended time window where the specimens can still be used. The study can simulate an extended stop of a measurement laboratory where it is crucial to know whether or not a limited number of specimens can still be used for DIC measurements.

### 2.2.9. The GOM ARAMIS

The DIC systems used in the research were the GOM ARAMIS 5M measuring system. The ARAMIS systems are non-contact and material-independent measuring systems that operate on the principle of DIC. They provide a reliable solution for full-surface and point-based inspections, whether the object under testing is a few millimeters or several meters in size. Therefore, the ARAMIS range of sensors is suitable for the dynamic measurement of 3D coordinates, 3D displacement, and 3D surface deformation. In addition, material characterizations can be performed, including FLC (forming limit diagram) and tensile tests using DIC techniques.

The latest technology now allows high local resolution with two 12-megapixel cameras. Small deformation and local effects over large areas can be measured. Interchangeable camera brackets and pre-adjusted, manufacturer-certified measuring lenses make the system quick and easy to adapt to the area to be measured. Since the systems were designed for industrial use, they are stable, i.e., less sensor calibration is required. The maximum image acquisition speed of the ARAMIS 5M at full resolution is 25 fps. By reducing the width or height of the image, this value can be increased so that any misbehavior of components can be documented in detail as a function of time [39].

During the measurement, the task is to compare a series of images of a specimen coated with a unique pattern and to search for and trace identical points throughout the images. However, searching for a single pixel between two images may not yield results, as the color and/or intensity value of a pixel may be the same as that of several pixels in the

reference image. A solution is to identify a small area (facet, sub-frame—hereafter referred to as the evaluation window) in the vicinity of a pixel between images. The searched evaluation window is usually square, which can be clearly identified through the whole image sequence due to the unique pattern taken during the test specimen preparation [40].

There are several methods to find a pixel around a part of another image. The two most common are normalized cross-correlation and normalized least squares. However, to achieve sufficient accuracy, the deformation of the evaluation windows must be taken into account, for which some form of iteration is usually used (Figure 3). In addition, most DIC software uses some variation of the Levenberg–Marquardt algorithm for nonlinear optimization.

Using the correct pattern in DIC tests is one of the most essential factors in reducing measurement noise and improving the accuracy of measurement results. GOM systems attach a reference image to the ideal speckle pattern. When preparing the test specimens, care should be taken to ensure that the painted pattern is as close as possible to the pattern for the measurement range. A pattern of sufficient quality is required because during DIC, a small area of the image—the evaluation window—is monitored as the sample is deformed and compared with the reference image, aiming to get the best possible fit of the evaluation window to the end of the measurement (see Figure 2) [52].

It is calculated from continuous monitoring of the change in grey levels and the match between images. As described above and based on previous pre-experiments, test specimens were measured with a suitable pattern for the measurements. However, this research does not focus on the patterns, as it focuses on the primer paint layer underneath the patterns.

## 3. Results and Discussion

This chapter presents the results of the tests presented in the measurement procedures. The painting/cleaning procedures which performed well and poorly on each test are highlighted for each test.

### 3.1. Wet Paint Film Thickness Measurement

Table 5 summarizes the results of the wet paint film thickness measurements, which clearly show that all paints cover the base plate in almost the same layer, with only the P11 sample showing significant differences. Paint which is too thick is dangerous because the layers may crack during drying. There were no differences between the different cleaning methods and base materials used in the test, so they are not shown in the table.

**Table 5.** Results of the wet paint thickness measurement.

| ID. | Name | Paint Thickness [μm] | | |
|---|---|---|---|---|
| | | Lower Limit | Upper Limit | Average |
| P1 | Acrylic Primer | 100 | 125 | 112.5 |
| P2 | Acrylic Mat Primer | 100 | 125 | 112.5 |
| P3 | High Temp. Paint | 75 | 100 | 87.5 |
| P4 | Heat Resistant | 125 | 150 | 137.5 |
| P5 | Spray Putty | 125 | 150 | 137.5 |
| P6 | Acryl | 125 | 150 | 137.5 |
| P7 | Acryl | 125 | 150 | 137.5 |
| P8 | Radiator | 50 | 75 | 62.5 |
| P9 | Aqua Eco+ | 75 | 100 | 87.5 |
| P10 | Chalk Finish Broken white | 125 | 150 | 137.5 |
| P11 | HB Body 950 | 600 | 650 | 625 |

### 3.2. Touch Test

In the touch test, two cleaning agents (SC6 and SC10) were excluded because they reacted with the paints on the different steel sheet materials and gave a bubbly, mottled

finish. These surface defects are not acceptable for a DIC measurement, so these cleaning procedures were no longer used for further testing. Table 6 shows the results of the test, which include the time in minutes for the print-free test. The faster-drying paints dry between 15 and 35 min; the P9 paint dries in 45 min.

**Table 6.** Results of the touch test.

| ID. | Name | Drying Time [min] |
|-----|------|-------------------|
| P1 | Acrylic Primer | 35 |
| P2 | Acrylic Mat Primer | 35 |
| P3 | High Temp. Paint | 35 |
| P4 | Heat Resistant | 35 |
| P5 | Spray Putty | 15 |
| P6 | Acryl | 15 |
| P7 | Acryl | 15 |
| P8 | Radiator | 20 |
| P9 | Aqua Eco+ | 45 |
| P10 | Chalk Finish Broken white | 5 |
| P11 | HB Body 950 | 25 |

Figure 12 shows the surface defects described above, with SC6 leaving bubbles and SC10 leaving spots on the surface. There were no differences between the different cleaning methods and raw materials applied in the test, so they are not shown in the table.

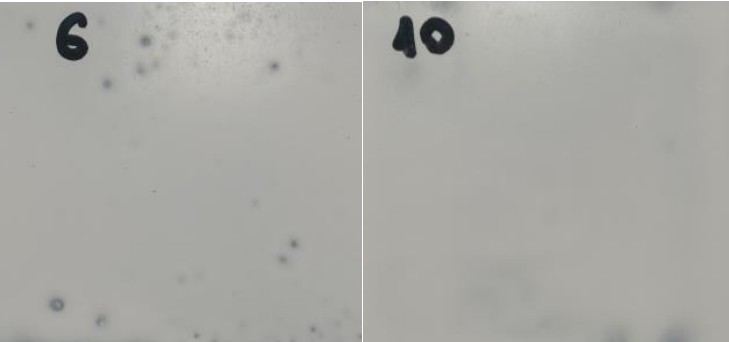

**Figure 12.** Photos of incorrect cleaning procedures.

*3.3. Print-Free Test*

The starting time for the print-free tests was the time of the results of the touch tests. After these times, the measurement windows were followed every 15 min. Table 7 shows the drying times for weights up to 1000 g.

**Table 7.** Print-free drying time.

| ID. | Name | Print-Free Drying Time [min] |
|-----|------|------------------------------|
| P1 | Acrylic Primer | 50 |
| P2 | Acrylic Mat Primer | 35 |
| P3 | High Temp. Paint | 30 |
| P4 | Heat Resistant | 35 |
| P5 | Spray Putty | 45 |
| P6 | Acryl | 35 |
| P7 | Acryl | 45 |
| P8 | Radiator | 35 |
| P9 | Aqua Eco+ | 155 |
| P10 | Chalk Finish Broken white | 20 |
| P11 | HB Body 950 | 70 |

Based on the results, no paints or residual cleaning processes were excluded. In addition, no differences were found between the different cleaning methods and raw materials used in the study and are therefore not included in the table.

### 3.4. Cross-Cut Test

The results of the cross-cut test are summarized in Table 8. Several conclusions can be drawn from the measurements. First, there was no difference in the measurement results between the material qualities; all samples S1–S5 behaved similarly. Therefore, it can be assumed that the substrate material or the sheet thickness does not play a role in the adhesion of the paint, only the surface quality, in which there was no difference between the samples. The test times also gave different results than expected. There was also no difference in results after 0 (immediate measurement after the time of touch drying), 6, 12, and 24 h. The samples rated "OK" performed well at the measurements' beginning and end. On the other hand, samples with a "NOK" rating failed at some measurements' stage and were excluded from the study. Based on the results, the dyes P1, P2, P6, P8, P9, and P11 did not pass and are therefore excluded from further testing. Furthermore, among the cleaning agents, SC3, SC7, and SC11 were excluded as they did not ensure good adhesion to the base plate, even for paints that achieved good ratings.

**Table 8.** Results of the cross-cut test.

| | | Materials: S1–S5 | | | | | | | | | |
|---|---|---|---|---|---|---|---|---|---|---|---|
| | | Drying Time: 0, 6, 12, 24 h | | | | | | | | | |
| ID. | Name | SC0 | SC1 | SC2 | SC3 | SC4 | SC5 | SC7 | SC8 | SC9 | SC11 |
| P1 | Acrylic Primer | | | | | | NOK | | | | |
| P2 | Acrylic Mat Primer | | | | | | NOK | | | | |
| P3 | High Temp. Paint | | | | | | OK | | | | |
| P4 | Heat Resistant | | | | | | OK | | | | |
| P5 | Spray Putty | | | | | | OK | | | | |
| P6 | Acryl | | | | | | NOK | | | | |
| P7 | Acryl | | | | | | OK | | | | |
| P8 | Radiator | | | | | | NOK | | | | |
| P9 | Aqua Eco+ | NOK | OK | | NOK | | OK | | | NOK | OK |
| P10 | Chalk Finish Broken white | | OK | NOK | | OK | | NOK | OK | NOK | OK |
| P11 | HB Body 950 | | | | | | NOK | | | | |

From the results, it can be stated that with the correct paint/cleaning procedure, adequate adhesion can be achieved regardless of the quality of the sheet and within 24 h after the time of the print-free test. Formability tests were carried out with the remaining combinations in the following tests.

### 3.5. Bending Test

During the bending test, it was found that the P5 and P10 paints were poorly milky with the application of all the remaining cleaners (Table 9). In addition, the bending test combined with scratching showed that the paint along the cut did not follow the plate deformation and separated from it, as shown in Figure 13, which shows micrographs of the specimens.

### 3.6. Erichsen Test Results by Visual Inspection

The Erichsen test was first used to detect any visible staining. Then, the measurement simulates the deformations expected during plate tests or plate forming processes well, so that a sound conclusion can be drawn about the paint–plate relationships. Table 10 shows the results and clearly shows that the remaining three paints (P3, P4, P7) tested well with all seven cleaning agents (SC0, SC1, SC2, SC4, SC5, SC8, SC9).

Figure 13 clearly shows the paint peeling off, while Figure 14 shows the successful paintings. Here, the contact between the paint and the base plate is stable along the cut.

**Table 9.** Results of the bending test.

| ID. | Name | Materials: S1–S5 | | | | | | |
|---|---|---|---|---|---|---|---|---|
| | | Drying Time: 0, 6, 12, 24 h | | | | | | |
| | | SC0 | SC1 | SC2 | SC4 | SC5 | SC8 | SC9 |
| P3 | High Temp. Paint | | | | OK | | | |
| P4 | Heat Resistant | | | | OK | | | |
| P5 | Spray Putty | | | | NOK | | | |
| P7 | Acryl | | | | OK | | | |
| P10 | Chalk Finish Broken white | | | | NOK | | | |

**Table 10.** Results of the visual inspection of the Erichsen test.

| ID. | Name | Materials: S1–S5 | | | | | | |
|---|---|---|---|---|---|---|---|---|
| | | Drying Time: 0, 6, 12, 24 h | | | | | | |
| | | SC0 | SC1 | SC2 | SC4 | SC5 | SC8 | SC9 |
| P3 | High Temp. Paint | | | | OK | | | |
| P4 | Heat Resistant | | | | OK | | | |
| P7 | Acryl | | | | OK | | | |

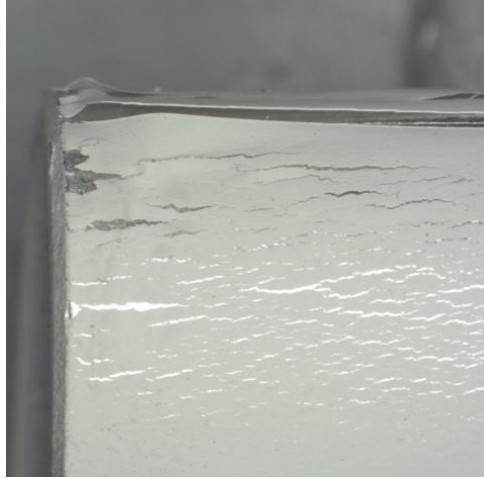 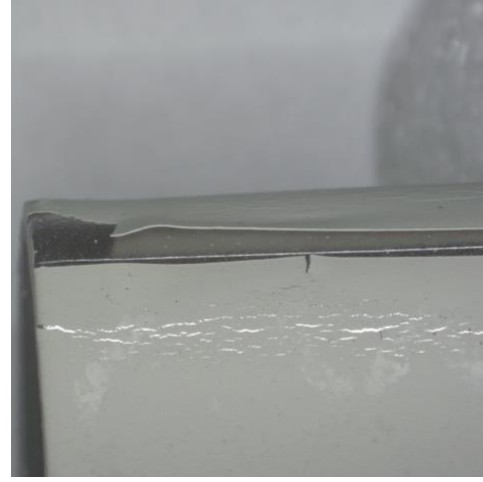

**Figure 13.** Microscope images of P5 and P10 paints.

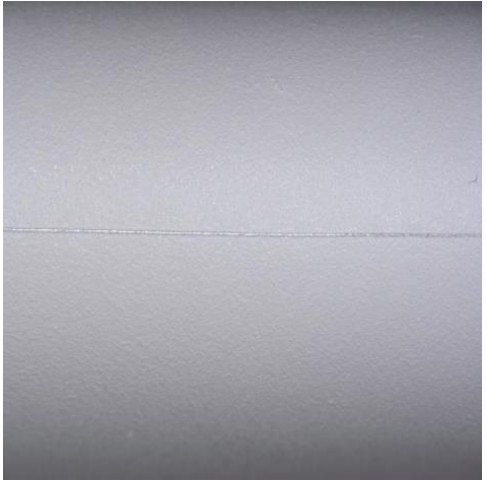 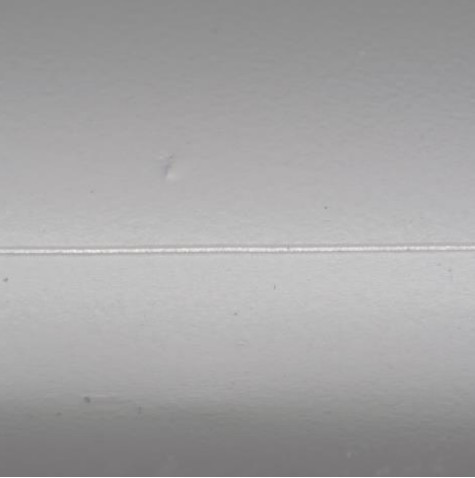

**Figure 14.** Results of a successful bending test.

### 3.7. Results of DIC Tests, Comparison of Thickness Reductions

In this research phase, Erichsen tests were performed using DIC recording with the remaining paint/cleaning procedures. In the first step of the tests, the paints were visually qualified, as described in Section 2.2.6. After the qualification, the DIC evaluation followed, where the thickness reduction (–eps3) values were determined for each specimen.

The visual inspection showed that all sample types received a satisfactory (1) rating within 24 h using three paintings (P3, P4, P7) and seven cleaning procedures (SC0, SC1, SC2, SC4, SC5, SC8, SC9). Therefore, no difference was observed in the rating, and the analysis continued with the DIC evaluation.

Figure 15 shows the S2 steel plate specimen with the GOM ARAMIS evaluation for the P7/SC2 painting and cleaning procedure, while the values of plate thickness reduction along the specimen's longitudinal section are illustrated in Figure 16.

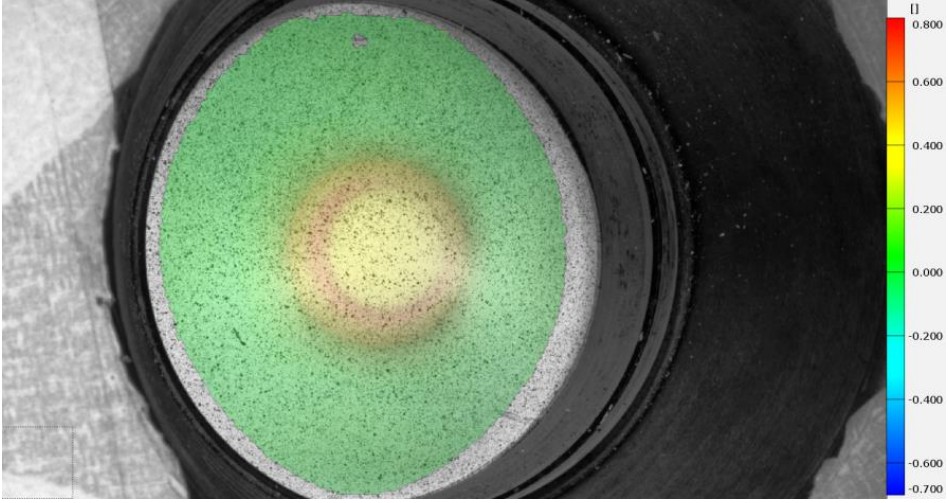

**Figure 15.** S2 sample plate with P7/SC2 painting and cleaning process (the legend, i.e., the color map, gives the deformation in mm unit).

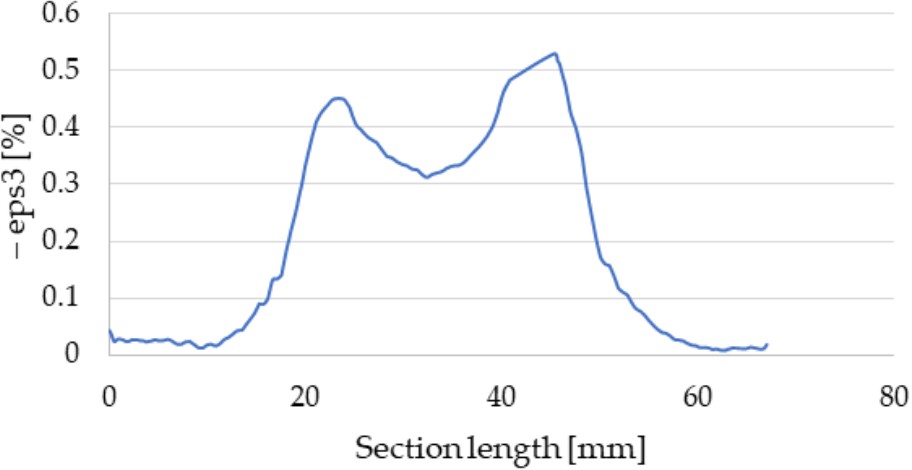

**Figure 16.** S2 Thickness reduction values with P7/SC2 combination.

This evaluation method was performed for each specimen for each painting and cleaning procedure.

To further investigate the performance of the paintings, the thickness reduction values at the peak were collected from each sample, or more precisely, their average. The following table (Table 11) summarizes these results.

**Table 11.** Thickness reduction results measured at the apex of the test specimen using different painting/cleaning procedures.

| ID. | Name | Materials: AVG S4 and S5 | | | | | | |
|-----|------|-----|-----|-----|-----|-----|-----|-----|
| | | Thickness Reduction: −eps3 [%] | | | | | | |
| | | SC0 | SC1 | SC2 | SC4 | SC5 | SC8 | SC9 |
| P3 | High Temp. Paint | 0.306 | 0.290 | 0.340 | 0.258 | 0.341 | 0.293 | 0.332 |
| P4 | Heat Resistant | 0.314 | 0.324 | 0.271 | 0.254 | 0.217 | 0.237 | 0.222 |
| P7 | Acryl | 0.249 | 0.266 | 0.256 | 0.271 | 0.321 | 0.249 | 0.355 |

The GOM ARAMIS determines the engineering elongations during the DIC measurement; the software writes out the −eps3 values when calculating the thickness reduction. Therefore, to be able to compare the thickness in mm units, it is necessary to establish a relationship between the engineering and real elongations. This relationship is given by the following formulas below.

$$engineering\ elongation:\ \varepsilon = \frac{l_1 - l_0}{l_0} \tag{1}$$

$$logarithm\ elongation:\ \varphi = ln\frac{l_1}{l_0} \tag{2}$$

$$relationship:\ \varphi = \ln(1 + \varepsilon) \tag{3}$$

Parameters in the above relationships (1, 2, 3) have the following meanings: $\varepsilon$—engineering elongation [%], $l_1$—elongated length of the specimen [mm], $l_0$—undeformed length of the specimen [mm], and $\varphi$—logarithm elongation [%].

After transforming, the plate thickness changes, and the following relationship can be obtained, from which the exact thickness value can be calculated from the ARAMIS measurement results.

$$\varphi_s = ln\frac{s_1}{s_0} \tag{4}$$

$$expressed\ in:\ s_1 = s_0 \cdot e^{\varphi_s} \tag{5}$$

Using this relationship (Equation (5)), the measured results are now shown together with the calculated results in Table 12. The meanings of the parameters in the above relations (Equations (4) and (5)) are: $\varphi_s$—logarithm elongation in thickness direction [%], $s_0$—initial plate thickness [mm], $s_1$—thinned plate thickness [mm], and $e$ is the Euler-number.

**Table 12.** Comparison of ARAMIS and thickness measurement results.

| ID. | Name | Measuring Device | Materials: S4 | | | | | | |
|-----|------|------------------|-----|-----|-----|-----|-----|-----|-----|
| | | | Thickness Reduction: [mm] | | | | | | |
| | | | SC0 | SC1 | SC2 | SC4 | SC5 | SC8 | SC9 |
| P3 | High Temp. Paint | ARAMIS | 0.76 | 0.81 | 0.75 | 0.70 | 0.75 | 0.80 | 0.78 |
| | | Gauge | 0.72 | 0.79 | 0.71 | 0.72 | 0.80 | 0.69 | 0.83 |
| P4 | Heat Resistant | ARAMIS | 0.72 | 0.71 | 0.76 | 0.78 | 0.81 | 0.79 | 0.81 |
| | | Gauge | 0.77 | 0.79 | 0.70 | 0.71 | 0.69 | 0.70 | 0.69 |
| P7 | Acryl | ARAMIS | 0.78 | 0.76 | 0.77 | 0.76 | 0.71 | 0.78 | 0.67 |
| | | Gauge | 0.72 | 0.73 | 0.72 | 0.70 | 0.83 | 0.69 | 0.82 |

Table 12 compares the values of the thickness reduction at the top of the peak, extracted from ARAMIS, and the results of the hourly measurements. These results can be used to qualify the painting/cleaning process because if the results are close to each other, the process under investigation is suitable for the DIC technology.

From the results, it can be seen that the deviations are already above the measurement error limit (0.02 mm) of the dial gauge (presented in Section 2.2.7), but within 5%, which

is still an acceptable result for a DIC measurement, so it can be stated that the paints and cleaning procedures are good at conveying the deformation of the plates.

The research also observed that thicker stainless and corrosion-resistant plates (S1) with a surface quality corresponding to a polished or well-ground surface quality (Ra 0.005–0.05, ISO 2768 [53]) performed worse in this test. However, the measurement results do not give as significant a deviation as the aluminum sheets with similar surface quality.

The results in Table 13 show more significant deviations of over 15–20%.

**Table 13.** Measurement results of usability tests.

| | | | Materials: S1 | | | | | | |
|---|---|---|---|---|---|---|---|---|---|
| | | | **Thickness Reduction: [mm]** | | | | | | |
| **ID.** | **Name** | **Measuring Device** | **SC0** | **SC1** | **SC2** | **SC4** | **SC5** | **SC8** | **SC9** |
| P3 | High Temp. Paint | ARAMIS | 1.42 | 1.45 | 1.35 | 1.51 | 1.35 | 1.44 | 1.37 |
| | | Gauge | 1.66 | 1.64 | 1.64 | 1.70 | 1.66 | 1.70 | 1.62 |
| P4 | Heat Resistant | ARAMIS | 1.45 | 1.46 | 1.39 | 1.56 | 1.41 | 1.46 | 1.41 |
| | | Gauge | 1.68 | 1.65 | 1.67 | 1.71 | 1.63 | 1.72 | 1.64 |
| P7 | Acryl | ARAMIS | 1.45 | 1.51 | 1.39 | 1.52 | 1.37 | 1.43 | 1.39 |
| | | Gauge | 1.65 | 1.68 | 1.65 | 1.69 | 1.71 | 1.73 | 1.65 |

However, the other samples, which typically have a mat and "rougher" surface, show similar differences to Table 12. Therefore, it can be concluded that for steels with outstanding surface quality, DIC measurement results can show a variation of around 10%; depending on the application, it should be considered whether or not the results obtained can be used.

*3.8. Durability Test*

A longer-term usability test was conducted to complement the results of the DIC study. For the tests, the steel sheet material S4 was selected with P3, P4, and P7 paints, as these paints performed the best based on previous measurements. For the selection of the cleaning agents, three of the seven types that remained in the tank were chosen (SC1, SC4, SC9), which were the most readily available and industrially available. The Erichsen tests were measured after a waiting period of 24 h for one week and again after two weeks. The study aimed to determine the more extended time window where the specimens can still be used. The study can simulate an extended stop of a measurement laboratory where it is crucial to know whether or not a limited number of specimens can still be used for DIC measurements.

Table 14 shows the averaged thickness measurement results, including hourly and ARAMIS measurements. The results show that P3, P4, and P7 paints still behaved well after one week with SC1 and SC4 cleaners for all S samples. Paint P7 with the SC9 cleaner on S samples was also replaced after one week. After two weeks, only the P3 paint passed the tests; the other dyes did not. Table 15 shows the rating results calculated from the test results, where deviations of less than 10% are adequate; those above can no longer be used safely to analyze plate formability processes. Although the qualification value of 10% was determined based on preliminary experiments, this error limit is also the measurement process error, above which the acceptance of the measurement results is limited in practice.

Figure 17 shows the S4 sample painted with P7 and P4 and prepared with SC4 and SC1 cleaner. Again, the paint pick-up and flaking can be observed, which makes the DIC measurement completely unevaluable.

As seen from the results in Table 15, P3 can be used for up to two weeks with SC1, SC4, and SC9 cleaning agents but not the other paints.

The following correlations and conclusions can be drawn based on the experiments. At the end of the research, it was seen that seven out of the twelve cleaning agents (SC0, SC1, SC2, SC4, SC5, SC8, SC9) gave satisfactory results for all five materials. Cleaners

SC6 and SC10 were excluded at the beginning of the tests because they caused a blotchy, bubbly surface for all the dyes.

**Table 14.** Measurement results of durability tests.

| | | Materials: S4 | | | | | |
| | | Drying Time: 1 Week | | | | | |
| | | SC1 | | SC4 | | SC9 | |
| ID. | Name | Thickness from Gauge (AVG) | Thickness from ARAMIS (AVG) | Thickness from Gauge (AVG) | Thickness from ARAMIS (AVG) | Thickness from Gauge (AVG) | Thickness from ARAMIS (AVG) |
|---|---|---|---|---|---|---|---|
| P3 | High Temp. Paint | 0.69 | 0.67 | 0.71 | 0.64 | 0.73 | 0.66 |
| P4 | Heat Resistant | 0.72 | 0.67 | 0.71 | 0.66 | 0.70 | 0.68 |
| P7 | Acryl | 0.74 | 0.66 | 0.75 | 0.67 | separated | – |
| | | Materials: S4 | | | | | |
| | | Drying time: 2 weeks | | | | | |
| P3 | High Temp. Paint | 0.70 | 0.65 | 0.71 | 0.67 | 0.70 | 0.68 |
| P4 | Heat Resistant | separated | – | separated | – | separated | – |
| P7 | Acryl | separated | – | separated | – | separated | – |

**Table 15.** Certification results of usability tests.

| | | Materials: S1–S5 | | |
| | | Drying Time: 1 week | | |
| ID. | Name | SC1 | SC4 | SC9 |
|---|---|---|---|---|
| P3 | High Temp. Paint | OK (thickness reduction difference under 10%) | | |
| P4 | Heat Resistant | OK (thickness reduction difference under 10%) | | |
| P7 | Acryl | OK (thickness red. diff. under 10%) | | NOK |
| | | Materials: S1–S5 | | |
| | | Drying time: 2 weeks | | |
| P3 | High Temp. Paint | OK (thickness reduction difference under 10%) | | |
| P4 | Heat Resistant | NOK (the paint separated) | | |
| P7 | Acryl | NOK (the paint separated) | | |

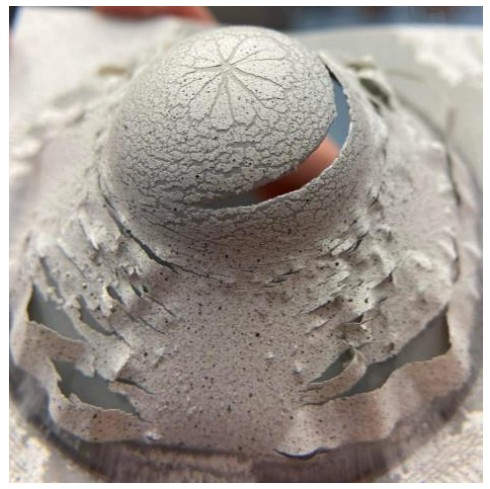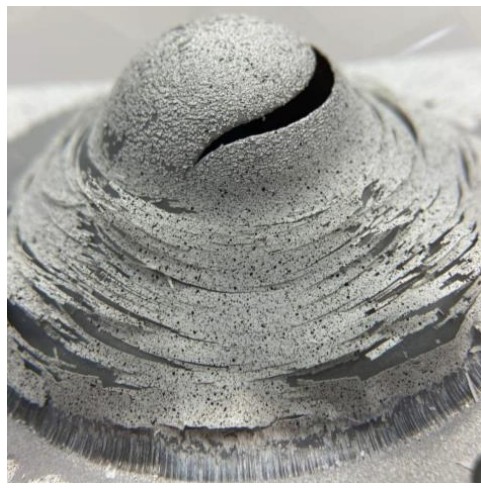

**Figure 17.** Separated paint after forming as shown for P7/SC4, P4/SC1.

One of the most essential criteria for DIC measurements is that the random speckle pattern and the base color are contrasting and well separated. This condition cannot be fulfilled by the painting and bubbling primer, and these two agents were excluded. Following the drying tests, which give the drying times after which the test specimen can be moved, the grid cut tests were carried out. The results of the grid cut or scratch test are of paramount importance, as this standard test is used to determine whether the paint adheres well to the surface of the plate.

The P1, P2, P6, P8, P9, and P11 paints on the tested S1, S2, S3, S4, and S5 steel materials did not show good results with any surface preparation procedures within the 24 h tested. Also, at this test stage, SC3, SC7, and SC11 cleaners were excluded as none showed good adhesion properties with any substrates or paints. Evaluating the results, it can be seen that it does matter which dye is selected during the DIC preparation phase of steel sheet materials, as the wrong dye will not give good test results. It is also confirmed that the surface preparation method must also be well chosen because even a good paint will not convey good deformation information in case of a wrong material choice.

Formability tests followed the paint adhesion tests. Pre-testing of the bending tests found that the remaining test combinations did not crack the surface, looking like they had adequate adhesion. However, when combining bending with scratching, it was already apparent that not all painting/cleaning processes have good adhesion. This test is critical because most plate qualification processes last until the specimen cracks, and it is important that the pattern remains correct even after cracking, i.e., adheres well. Based on the measurements, P5 and P10 paints were found to have separated from all steel materials with the remaining cleaners and were therefore excluded.

Erichsen tests have two functions in research—validation and qualification. The Erichsen tests were first visually verified, where P3, P4, and P7 paints with cleaners SC0, SC1, SC2, SC4, SC5, SC8, and SC9 passed on all five materials within the first 24 h. Subsequently, after evaluation of the ARAMIS tests, the plate thickness reduction results measured at the post-tensile specimen peak were extracted from the DIC results and compared with the results of the gauge thickness measurements. The following conclusions can be formulated from the comparison of the results.

For the samples with a typical mat surface or a mat surface coating (S2, S3, S4, S5), the remaining paint/cleaner pairings performed well, with deviations within the 10% margin of error. However, for sample S1, more than 15–20% of deviations occurred within the 24 h test limit. This sample differs from the others in surface quality, as it is almost polished (Ra 0.005–0.05) with a shiny surface.

The measurements support the hypothesis that DIC measurements require selecting the appropriate painting/cleaning procedure and even the appropriate procedure for the material and surface quality to obtain results with sufficient accuracy. This assumption is confirmed by the results of the durability tests, where sample S1 performed poorly in all test aspects.

The durability test showed that after one week of drying time, P3, P4, and P7 paints passed with SC1 and SC4 cleaners with less than 10% deviation, but P7/SC9 samples failed, unlike P3 and P4 paints. However, after two weeks, P3 paint passed the tests with SC1, SC4, and SC9 cleaners. These experiences may be helpful for long-term DIC measurements where the paint is vital to remain stable but flexible for several days or weeks.

## 4. Conclusions

First of all, the authors must note that the manufacturers (neither for detergents, i.e., the cleaning agents, nor for paints) do not give an exact, usable chemical composition; they only list the ingredients. Therefore, providing results and conclusions that can be supported scientifically and stand up to interpretation is not easy. It was the reason why the authors did not address this in the recording and formulation of the conclusions.

Because of the fact that the authors have more than 10 years of experience in DIC measurements, the preparation of adequate, appropriate, and the same painting technology, and also the same speckle pattern for all the specimens, was guaranteed.

Based on the above paragraph and the experiments performed, the research results show a significant difference between commercially available spray paints. Based on the samples tested, many of them are unsuitable. Furthermore, it has been shown that all the surface preparation and cleaning products—tested in this study—cannot be applied for measurements combined with the 3D-DIC method because the paint cannot adhere properly to the surface. Therefore, the authors aimed to examine, in a detailed manner, the considered twelve types of cleaning agents (SC0–SC11) and the eleven types of spray paints (P1–P11) on five different types of steel materials (S1–S5). In the cases of steel plates (sheets), the thicknesses vary between 0.75 to 2.00 mm.

The principal goal was to investigate the $12 \times 11 \times 5 = 660$ combinations of the chosen and considered cleaning agents, spray paints, and steel materials. The experiments were performed by wet paint film thickness tests, touch tests, print-free tests, cross-cut tests, bending tests, Erichsen tests with DIC evaluation and traditional dial gauge measurement, and durability tests.

The order of the measurements was the order mentioned above. In the case a given combination failed at a given test, it was not taken into consideration in the next step.

It can be concluded that for the samples with a typical mat surface or a mat surface coating (i.e., steel materials S2, S3, S4, S5), in the end, remaining paint/cleaning agent pairings performed well, with deviations within the 10% margin of error. However, for sample S1, more than 15–20% deviations occurred within the 24-h test limit. It must be noted that this steel material has a polished (Ra 0.005–0.05) and shiny surface.

The measurements support the hypothesis that DIC measurements require selecting the appropriate painting/cleaning procedure and even the appropriate procedure for the material and surface quality to obtain results with sufficient accuracy. This assumption is confirmed by the results of the durability tests, where sample S1 performed poorly in all test aspects.

The durability test showed that after one week of drying time, the P3, P4, and P7 paints passed with SC1 and SC4 cleaners with less than 10% deviation, but P7/SC9 samples failed, unlike P3 and P4 paints. However, after two weeks, the P3 paint passed the tests with SC1, SC4, and SC9 cleaners. These experiences may be helpful for long-term DIC measurements where the paint is essential to remain stable but flexible for several days or weeks.

To be able to draw final conclusions, it was determined that for steel plates, P3, P4, and P7 paints performed best with SC1 and SC4 cleaning agents, and it is recommended that these combinations be used for DIC tests. Furthermore, it was found that these paints can be used for up to 1 week for such measurements.

As a future research direction, the authors suggest obtaining larger quantities of steel sheets, cleaning agents, and paint sprays from the same different production batches and storing them in different but strictly controlled ways. By this, it is meant that they would be exposed to different temperatures, humidity, UV radiation, etc. On this basis, it should be investigated to what extent and how these influence the behavior of the paints applied to the plates, for example, by a combination of the test series presented here and the Erichsen cupping test and DIC measurements. Furthermore, it could be an exciting investigation to subject the already-prepared painted plates to different effects. Nevertheless, of course, these would involve a significant number of combinations and would also present a severe challenge to researchers in terms of cost and measurement time.

**Author Contributions:** Conceptualization, S.S. and B.F.S.; methodology, S.S.; software, S.S. and B.F.S.; validation, S.S. and B.F.S.; formal analysis, S.S. and B.F.S.; investigation, S.S., B.F.S., D.K., A.N., M.S. and S.F.; resources, S.S., A.N. and S.F.; data curation, S.S. and B.F.S.; writing—original draft preparation, S.S., B.F.S., D.K., A.N., M.S. and S.F.; writing—review and editing, S.S., B.F.S., D.K., A.N., M.S. and S.F.; visualization, S.S. and B.F.S.; supervision, D.K., A.N., M.S. and S.F.; project administration, S.S.; funding acquisition, S.F. All authors have read and agreed to the published version of the manuscript.

**Funding:** This research received no external funding.

**Data Availability Statement:** Not applicable.

**Acknowledgments:** This paper was prepared by the research team "SZE-RAIL".

**Conflicts of Interest:** The authors declare no conflict of interest.

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
