# Peer review of "Optimization of Surface Preparation and Painting Processes for Railway and Automotive Steel Sheets"

_infrastructures, doi:10.3390/infrastructures8020028_

Round 1
Reviewer 1 Report
The paper deals with the optimization of surface preparation and painting processes for railway and automotive steel sheets. The authors applied standard methodologies (wet painting thickness test, touch test, print-free test, cross-cut test, bending test, as well as Erichsen cupping test) for the evaluation of painting methods related to the possibility of applying DIC (Digital Image Correlation) measurement methodology. Twelve cleaning agents, eleven painting materials (sprays), and five steel base materials/sheets/plates have been considered for the laboratory tests. The paper demonstrates the importance of the preparatory work for GOM ARAMIS measurement because the quality of the base-paint and the speckle pattern mainly affect the quality and reliability of the laboratory tests executed and assessed by optimal measurement system. In the case the paint (or painting) has inadequate quality, the measured deformation values are incorrect; which would lead to inappropriate, i.e. inaccurate results. The bonding and drying of the painting depend on the cleaning agents used, as well as the painting itself. There is another main factor: the type of the base plate. The paper gives up-to-date and interesting results for this problem.
The paper matches the scope of Infrastructures. It is well written and structured. Objectives are adequately defined and addressed. The quality of presentation is adequate.
The reviewer recommends the below-mentioned modifications:
• In numbering subsections, 2.2.1 is repeating, so this must be corrected. The authors may also consider to put all 2.2.x subsections into a new general Section 3 - only a suggestion.
• Please add doi number where missing, e.g., ref. 23, ref. 28, refs. 34-41, ref. 43, etc. and make sure all the references are strictly formatted as per journal template.
• Another suggestion: a list of abbreviations at the end of the paper might be helpful to the reader.
• In many places, references have been introduced after the fullstop, i.e. after the sentence is finished, which is not correct. This needs to be corrected.
• The introduction part can be improved with more relevant and up-to-date papers. The authors should appropriately extend this section by discussing more relevant works focusing on different methods and models in the literature.
• Why have you chosen the given types of cleaning agents, sprays and steel materials for the tests? Any specific reason behind this choice?
• The conclusions should include some outlook of the future work in the field.
Author Response
See the attached PDF file.

Reviewer 2 Report
1. Although the experiments were well designed and paper is nicely written, I think this paper can do better in scientific soundness. This paper covers a series of experiments with different paint and cleansing agents and finally presents the best of them but it does not provide the reasoning behind these results. Without the explanation of results, conclusion is merely a summary of results. You should try to explain, why particular specimen, paint and surface cleaners performed better than others did. Can this be explained by the chemical composition of the paint and surface cleaners?
2. If possible, it will be helpful for the reader if model numbers of paints and surface cleaners are included in the paper.
Author Response
See the attached PDF file.

Reviewer 3 Report
The paper investigates commercially available surface preparation and cleaning materials and spray paints used in digital image correlation for railway and automotive steel plates. Twelve surface cleaners, eleven paints, and five steel sheet materials were tested. The main value of the work lies in determining the optimal combinations of cleaning agents and paints for DIC studies of steel plates. Therefore, the article should be accepted after the following major and minor points are considered and addressed by the authors.
Major points:
1. The surface cleaning and painting procedures are used for the DIC calculations. Although the focus of this study is on the primer paint layer underneath the patterns, different speckle patterns and DIC parameters will influence the DIC results. Authors need to explain how the same DIC paraments including speckle patterns are ensured in experiments for different specimens with different surface cleaning and painting procedures.
2. To compare the different surface cleaning and painting procedures, the measurement accuracy of DIC should be first determined. For example, the noise (or so-called measurement uncertainty) analysis of the displacement and strain fields obtained from taking multiple static images before deformation can be provided to verify the accuracy of the DIC results. And for DIC calculation, subset size, step size, and strain calculation window are important parameters affecting the DIC results. The choices of these DIC parameters should be briefly given.
Minor points:
1. It needs to be explained how this conclusion was drawn in the last sentence of the Abstract: Based on the results, DIC measurements can be performed with higher accuracy and safety in laboratory and industrial conditions.
2. Some minor textual adjustments are required. For examples, “2D-DIC”in line 78, but “2D DIC” in line 79; ”3D technique” in Lines 87 and 90 is actually ”3D-DIC technique”;“digital Image correlation”in line 162 can directly be“DIC”.
3. In the descriptions of different experiments, only the parameters of the steel sheet material and the testing equipment for each test are available, there are no pictures of the relevant specimens. The result discussions are also basically textual and tabular, more pictures of the experimental results would have been more convincing. Especially for the Erichsen tests with DIC, more pictures like Figs. 15 and 17 could be provided.
4. The colour scale of the cloud map in Figure 15 needs to be fully labelled, including the name and unit of the parameter, e.g. displacement/mm.
5. Check this sentence in Lines 681-683 of“4.Conclusions“: In addition, it has been shown that all of the surface preparation and cleaning agents tested cannot be used, as the paint cannot create sufficient adhesion on the surface.
Author Response
See the attached PDF file.
